# Ballistic superconductivity and tunable $\pi$-junctions in InSb quantum wells

Chung Ting Ke[1,7], Christian M. Moehle[1,7], Folkert K. de Vries[1], Candice Thomas[2,3], Sara Metti[3,4], Charles R. Guinn[2], Ray Kallaher[3,5], Mario Lodari[1], Giordano Scappucci [1], Tiantian Wang[2,3], Rosa E. Diaz[3], Geoffrey C. Gardner[3,5], Michael J. Manfra [2,3,4,5,6] & Srijit Goswami [1]

Planar Josephson junctions (JJs) made in semiconductor quantum wells with large spin-orbit coupling are capable of hosting topological superconductivity. Indium antimonide (InSb) two-dimensional electron gases (2DEGs) are particularly suited for this due to their large Landé g-factor and high carrier mobility, however superconducting hybrids in these 2DEGs remain unexplored. Here we create JJs in high quality InSb 2DEGs and provide evidence of ballistic superconductivity over micron-scale lengths. A Zeeman field produces distinct revivals of the supercurrent in the junction, associated with a $0-\pi$ transition. We show that these transitions can be controlled by device design, and tuned in-situ using gates. A comparison between experiments and the theory of ballistic $\pi$-Josephson junctions gives excellent quantitative agreement. Our results therefore establish InSb quantum wells as a promising new material platform to study the interplay between superconductivity, spin-orbit interaction and magnetism.

[1] QuTech and Kavli Institute of Nanoscience, Delft University of Technology, 2600 GA Delft, The Netherlands. [2] Department of Physics and Astronomy, Purdue University, West Lafayette, IN 47907, USA. [3] Birck Nanotechnology Center, Purdue University, West Lafayette, IN 47907, USA. [4] School of Electrical and Computer Engineering, Purdue University, West Lafayette, IN 47907, USA. [5] Microsoft Quantum at Station Q Purdue, Purdue University, West Lafayette, IN 47907, USA. [6] School of Materials Engineering, Purdue University, West Lafayette, IN 47907, USA. [7] These authors contributed equally: Chung Ting Ke, Christian M. Moehle. Correspondence and requests for materials should be addressed to S.G. (email: S.Goswami@tudelft.nl)

Two-dimensional electron gases (2DEGs) coupled to superconductors offer the opportunity to explore a variety of quantum phenomena. These include the study of novel Josephson effects[1], superconducting correlations in quantum (spin) Hall systems[2–7], hybrid superconducting qubits[8,9], and emergent topological states in semiconductors with strong spin-orbit interaction (SOI)[10–13]. Topological superconductivity in such 2DEGs can be realized using planar Josephson junctions (JJs), where the combined effect of SOI and a Zeeman field is known to significantly alter the current-phase relation[14–16]. In particular, one expects a complete reversal of the supercurrent (i.e., a $\pi$–JJ)[17–19] when the Zeeman and Thouless energy of the system become comparable. It was shown recently that such a $0-\pi$ transition in a 2D system is in fact accompanied by a topological phase transition[12,13,20,21]. This, combined with the promise of creating scalable topological networks[22–24], provides a strong motivation to study induced superconductivity in 2DEGs.

Key requirements for the semiconductor include low disorder, large SOI and a sizable Landé g-factor, combined with the ability to grow it on the wafer scale. InSb satisfies all of these requirements[25–28] and has emerged as a prime material candidate for engineering topological superconductivity, as evident from nanowire-based systems[29,30]. However, despite significant progress in the growth of InSb 2DEGs[31,32], material challenges have prevented a systematic study of the superconducting proximity effect in these systems.

Here, we overcome these issues and reliably create JJs, thus providing evidence of induced superconductivity in high quality InSb quantum wells. The JJs support supercurrent transport over several microns and display clear signatures of ballistic superconductivity. Furthermore, we exploit the large g-factor and gate tunability of the junctions to control the current-phase relation, and drive transitions between the 0 and $\pi$-states. This control over the free energy landscape allows us to construct a phase diagram identifying these 0 and $\pi$-regions, in agreement with theory.

## Results

**Induced superconductivity in InSb 2DEGs.** The JJs are fabricated in an InSb 2DEG wafer grown by molecular beam epitaxy, with a nominal electron density $n = 2.7 \times 10^{11}$ cm$^{-2}$ and mobility $\mu \approx 150,000$ cm$^2$V$^{-1}$s$^{-1}$, which corresponds to a mean free path $l_e \approx 1.3$ µm. Figure 1a shows a cross-sectional illustration and scanning electron micrograph of a typical JJ. Following a wet etch of the 2DEG in selected areas, NbTiN is deposited to create side-

contacts to the 2DEG, thus defining a JJ of width $W$ and length $L$. Prior to sputtering NbTiN, an in-situ argon plasma cleaning of the exposed quantum well is performed in order to obtain good electrical contacts. A metal top-gate, deposited on a thin dielectric layer is used to modify the electron density in the JJ. Details of the device fabrication and wafer growth can be found in the Methods section.

The junctions are measured using a quasi-four terminal current-biased circuit (Fig. 1a) at a temperature of 50 mK. We observe a clear supercurrent branch with zero differential resistance, d$V$/d$I$, followed by a jump to the resistive branch at switching current, $I_s$. In small perpendicular magnetic fields, $B_z$, Fraunhofer-like interference patterns are observed, as seen in Fig. 1b. The magnitude of supercurrent is controlled using the gate (Fig. 1c). Lowering the gate voltage, $V_g$, leads to a reduction of the electron density in the 2DEG and therefore to a suppression of $I_s$ and an increase in the normal state resistance, $R_n$. In addition, we observe multiple Andreev reflections indicating an induced superconducting gap of 0.9 meV, and excess current measurements allow us to estimate transparencies in the range of 0.6–0.7 (representative data are provided in the Supplementary Note 2).

**Ballistic superconductivity.** Studying JJs of varying lengths ($L = 0.7$–4.7 µm), we gain insight into the transport regime. These devices fall in the long junction limit, since their lengths exceed the induced superconducting coherence length of around 500 nm (see Supplementary Note 2). In this limit the product of the critical current, $I_c$, and $R_n$ is proportional to the Thouless energy[33], $E_{Th} = \hbar v_F l_e / 2L^2$, where $v_F$ is the Fermi velocity in the 2DEG. Thus, for ballistic (diffusive) transport where $l_e = L$ ($l_e < L$), we expect $I_c R_n$ to scale as $1/L$ ($1/L^2$). In our experiments we measure $I_s$, but expect it to be close to $I_c$, since the Josephson energy ($\approx 20$ K) is significantly larger than the fridge temperature ($\approx 50$ mK). Figure 1d shows $I_s R_n$ for a set of JJs. We find a $1/L$ scaling (black dots) indicative of ballistic superconductivity, with deviations only for the longer ($L \geq 2.7$ µm) junctions. Such a $1/L$ dependence was predicted decades ago[34] but has only recently been experimentally observed over micron-scale lengths in clean graphene-based JJs[35,36]. To confirm the scaling arguments we also include data from a lower mobility wafer (see Supplementary Note 1) with $l_e \approx 0.5$ µm (red dots) and find a $1/L^2$ scaling, consistent with diffusive behavior. In the remainder of this work we focus on JJs fabricated on the high mobility wafer.

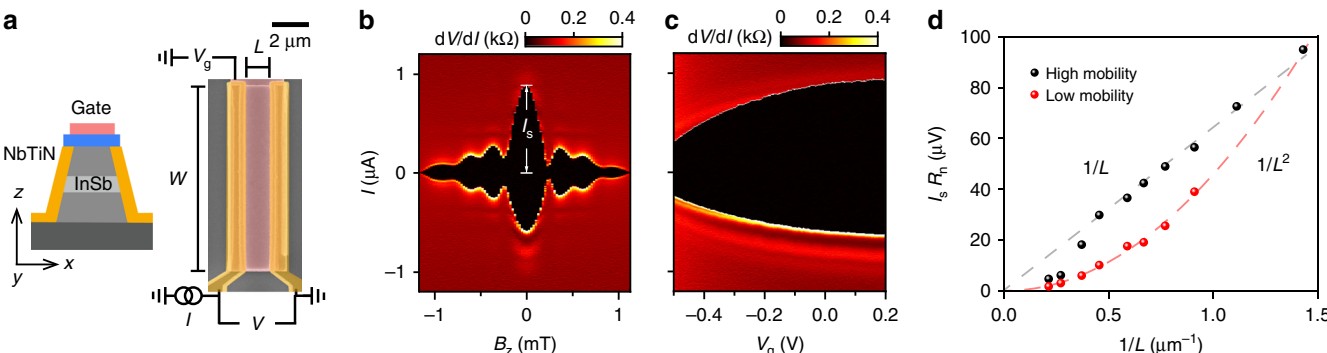

**Fig. 1** Ballistic superconductivity in InSb 2DEGs. **a** Cross-sectional schematic and false-colored scanning electron micrograph (along with a measurement schematic) of a top-gated JJ of width $W$ and length $L$. **b** Differential resistance, d$V$/d$I$, versus perpendicular magnetic field, $B_z$, and current bias, $I$, displaying a Fraunhofer-like interference pattern for a JJ with $W = 9.7$ µm, $L = 1.1$ µm. White line indicates the magnitude of the switching current, $I_s$, at zero magnetic field. **c** d$V$/d$I$ as a function of $I$ and gate voltage, $V_g$, for the same JJ, showing gate control of $I_s$. **d** Length dependence of $I_s R_n$ for JJs on a high mobility (black dots) and low mobility (red dots) wafer, obtained at $V_g = 0$ V. Dashed lines are $1/L$ and $1/L^2$ fits to the data, indicating ballistic and diffusive transport, respectively

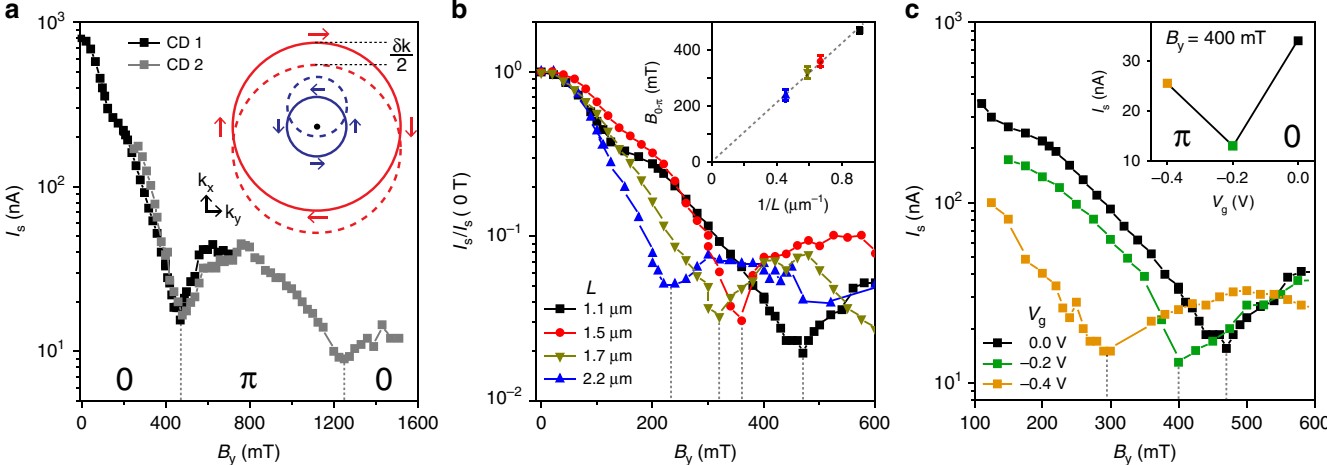

**Fig. 2** Magnetic field-driven $0-\pi$ transitions. **a** Variation of the switching current, $I_s$, with in-plane magnetic field, $B_y$, at $V_g = 0$ V for the same JJ as in Fig. 1b, c. Two distinct revivals of $I_s$ are visible at $B_y = 470$ mT and 1250 mT, associated with $0-\pi$ transitions. The data are from two cool downs (CDs). The momentum shift, $\delta k/2$, of the Fermi surfaces due to the Zeeman field is sketched in the inset. The solid (dashed) lines depict the situation at zero (finite) magnetic field, and the arrows represent the spin orientation. **b** $I_s$ as a function of $B_y$ at $V_g = 0$ V for four JJs with different lengths. For better visibility, $I_s$ is normalized with respect to $I_s$ at $B_y = 0$ T. Dashed lines indicate $B_{0\text{-}\pi}$, the field at which the transition occcurs for each length. The inset shows a linear dependence of $B_{0\text{-}\pi}$ on $1/L$, in agreement with ballistic transport. **c** $I_s$ vs. $B_y$ at three different $V_g$ for the JJ with $L = 1.1$ μm. $B_{0\text{-}\pi}$ shifts to lower values of $B_y$ with more negative gate voltages. $I_s$ vs. $V_g$ at $B_y = 400$ mT shows a non-monotonic behavior as displayed in the inset. The length and gate dependence of panel **b**, **c** are in qualitative agreement with Eq. (1)

**$0-\pi$ transitions in Josephson junctions.** Using these ballistic junctions, we now explore their response to a Zeeman field. The theory of JJs with large SOI subjected to a magnetic field has been discussed extensively[14,17,20]. Below we briefly describe the essential elements of the physical picture. At zero $B$ the Fermi surfaces are split due to the Rashba SOI (solid lines of Fig. 2a inset). The magnetic field then splits the bands by the Zeeman energy, $E_Z = g\mu_B B$, leading to a shift in the Fermi surfaces by $\pm\delta k/2$. The depicted shift of the Fermi surfaces assumes that the spin-orbit energy dominates over the Zeeman energy, which is indeed the case for the measured JJs (see Supplementary Note 3 for a detailed discussion). Therefore, Cooper pairs (electrons with opposite momentum and spin) now possess a finite momentum, given by $\mathbf{k}_F \cdot \delta\mathbf{k} = E_Z(m^*/\hbar^2)$, where $\mathbf{k}_F$ is the Fermi momentum and $m^*$ the effective mass. This translates to a phase acquired by the superconducting order parameter along the direction of current flow, $\Psi(\mathbf{r}) \propto \cos(\delta\mathbf{k} \cdot \mathbf{r})$[37–39]. Depending on the length of the Cooper pair trajectories, $|\mathbf{r}|$, the order parameter is either positive or negative, corresponding to the ground state of the JJ being at 0 or $\pi$ superconducting phase difference, respectively. This oscillation of the order parameter results in a modulation of the critical current $I_c \propto |\Psi|$, where a minimum of $I_c$ is expected whenever the order parameter switches sign[14,15]. Taking only trajectories perpendicular to the contacts ($\delta\mathbf{k} = \delta k\hat{\mathbf{x}}, \mathbf{k}_F = k_F\hat{\mathbf{x}}$), a JJ with length $L$ will display minima in $I_c$ when $L\delta k = (2N + 1)\pi/2$, with $N = 0, 1, 2...$. The condition for the first minimum ($N = 0$) can be expressed as a resonance condition in terms of the Zeeman and ballistic Thouless energy as $E_Z = \pi E_{Th}$ giving:

$$g\mu_B B = \pi\frac{\hbar^2\sqrt{2\pi n}}{m^* 2L}. \tag{1}$$

The $0-\pi$ transition therefore depends on three experimentally accessible parameters: (1) applied magnetic field, (2) length of the JJ, and (3) carrier density. In the following, we demonstrate independent control of each of these parameters, allowing for a complete study of the free energy landscape of the junctions.

**Magnetic field-driven $0-\pi$ transitions.** We start by varying $B_y$, while $n$ (controllable by $V_g$) and $L$ remain fixed. The orientation of the magnetic field reflects the Fermi surfaces described, and avoids unwanted geometric effects[40]. Figure 2a shows the expected oscillation of $I_s$ with increasing $B_y$, displaying two distinct minima at $B_y = 470$ mT and $B_y = 1250$ mT (see Supplementary Note 4 for details about magnetic field alignment). This behavior is consistent with a magnetic field-driven $0-\pi$ transition, as discussed above, where the first (second) minimum corresponds to a transition of the JJ state from 0 to $\pi$ ($\pi$–0). This interpretation is corroborated by the occurrence of the second minimum at a field value, which is approximately three times larger than the first. Note that this is incompatible with a Fraunhofer interference pattern that might arise from the finite thickness of the 2DEG. Furthermore, taking into account the gate dependence of the transition and other geometric considerations (discussed in detail in the Supplementary Note 5) allows us to conclusively rule out such a mechanism for the supercurrent modulation.

Next, we investigate how the length of the JJ influences $B_{0\text{-}\pi}$, the magnetic field at which the transition occurs. Figure 2b presents the $I_s$ oscillation for JJs with four different lengths, showing that $B_{0\text{-}\pi}$ is systematically reduced for increasing $L$. Plotting $B_{0\text{-}\pi}$ with respect to $1/L$ (inset of Fig. 2b), we find a linear dependence as expected from Eq. (1). The transition points are therefore determined by the ballistic $E_{Th}$, consistent with the conclusions from Fig. 1d. Finally, we check the dependence of the transition on the electron density. In Fig. 2c, we plot $I_s$ versus $B_y$ for different gate voltages using a JJ with $L = 1.1$ μm. As $V_g$ is lowered, $B_{0\text{-}\pi}$ shifts to smaller values, again in qualitative agreement with Eq. (1). Interestingly, above a certain magnetic field the state of the JJ (0 or $\pi$) becomes gate-dependent. For example at $B_y = 400$ mT, the junction changes from a 0-JJ ($V_g = 0$ V) to a $\pi$-JJ ($V_g = -0.4$ V), with a transition at $V_g = -0.2$ V. This indicates the feasibility of tuning the JJ into the $\pi$-state using gate voltages, while the magnetic field remains fixed.

**Gate-driven $0-\pi$ transitions.** These gate-driven transitions are demonstrated in Fig. 3a–d, which show a sequence of $I-V_g$ plots for increasing in-plane magnetic fields. At $B_y = 250$ mT, $I_s$ displays a monotonic reduction with decreasing $V_g$. At a higher

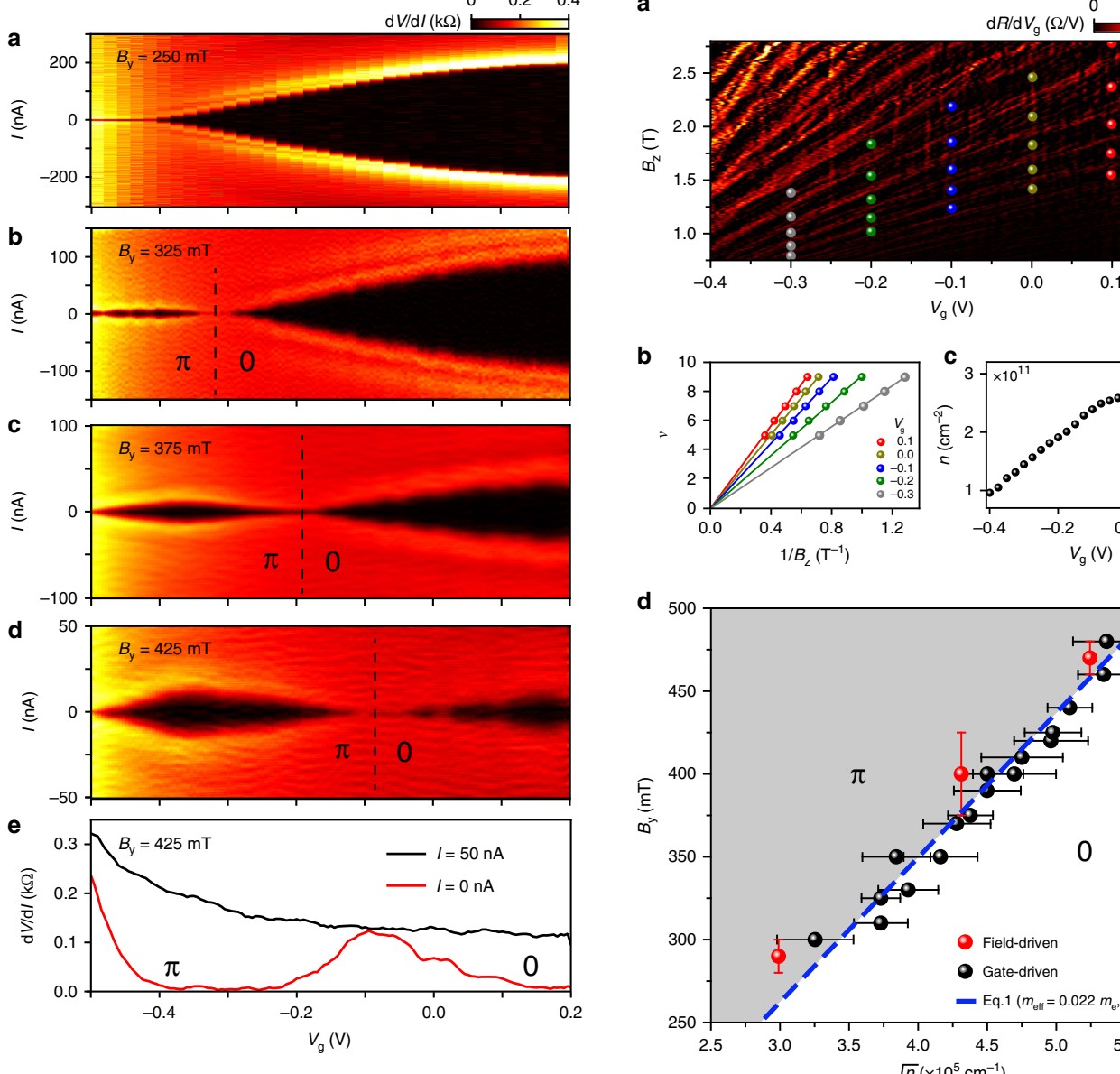

**Fig. 3** Gate-driven 0–$\pi$ transitions. **a**–**d** d$V$/d$I$ as a function of $I$ and $V_g$ for several $B_y$ as indicated. From $B_y = 325$ mT onward, a gate-driven 0–$\pi$ transition becomes evident, characterized by a re-emergence of $I_s$ with decreasing $V_g$. As expected, the transition shifts to higher gate voltages with increasing $B_y$ (see Supplementary Note 6 for sweeps at additional values of the magnetic field). **e** Line-cuts through panel **d** at $I = 50$ nA (black) and $I = 0$ nA (red). The low bias trace reveals the 0–$\pi$ transition whereas the high bias trace shows a monotonic behavior

magnetic field, $B_y = 325$ mT, $I_s$ reveals a markedly different behavior, whereby the supercurrent first decreases and then (at $V_g = -0.32$ V) shows a clear revival, indicative of a gate-driven 0–$\pi$ transition, where the resonance condition ($E_Z = \pi E_{Th}$) is achieved by tuning the electron density. Increasing $B_y$ further, continuously moves the transition point to higher gate voltages (larger density), perfectly in line with expectations for a 0–$\pi$ transition. Figure 3e shows two line-cuts from Fig. 3d. At zero current bias, d$V$/d$I$ shows a clear peak, indicative of a re-entrance of the supercurrent due to the the 0–$\pi$ transition. However, at high bias, d$V$/d$I$ increases monotonically, similar to the response at zero magnetic field. This eliminates trivial interference effects as an explanation for the supercurrent modulation, where one would expect a correlation between the two curves[35,41,42].

**Fig. 4** 0–$\pi$ phase diagram. **a** Landau fan diagram for the JJ with $L = 1.1\,\mu$m, showing the transresistance (d$R$/d$V_g$) as a function of $B_z$ and $V_g$. The symbols indicate positions of integer filling factors $\nu$ at specific values of $V_g$. **b** Dependence of $\nu$ on $1/B_z$ along with linear fits used to extract the electron density, $n(V_g)$, presented in **c**. **d** Phase diagram of the 0–$\pi$ transition as a function of $B_y \propto E_Z$ and $\sqrt{n} \propto E_{Th}$, containing all data points obtained from both field-driven (red) and gate-driven (black) 0–$\pi$ transitions. For the error analysis, see Supplementary Note 6. We fit the data to Eq. (1) (blue line) with $g_y$ as a fitting parameter

**Construction of the 0–$\pi$ phase diagram**. In contrast to the field-driven measurements (Fig. 2), controlling the transition with a gate avoids the need for time-consuming field alignment procedures, thus allowing us to efficiently explore a large parameter space in magnetic field and gate voltage. We now combine these results to construct a 0–$\pi$ phase diagram of the JJ. The combination of a high quality 2DEG and relatively long devices results in well defined magneto-resistance oscillations, allowing us to directly extract the electron density in the junction. Figure 4a shows the Landau fan diagram in perpendicular magnetic fields, $B_z$, from which we identify the filling factors, $\nu = nh/eB_z$ (Fig. 4b),

and thereby obtain the $n$ vs. $V_g$ curve (Fig. 4c). We then plot all the transition points in Fig. 4d. The axes represent the two important energy scales in the system ($B_y \propto E_Z$ and $\sqrt{n} \propto E_{Th}$), thereby highlighting the 0 and $\pi$ regions in the phase space. Finally, we compare our results with the theory of ballistic JJs represented by Eq. (1). To do so, we independently extract the effective mass (see Supplementary Note 7), $m^* = (0.022 \pm 0.002)$ $m_e$, and fit the data to a single free parameter, $g_y$ (the in-plane g-factor), giving $g_y = 25 \pm 3$ in good agreement with previous measurements on similar InSb quantum wells[28].

Our work provides the first evidence of induced superconductivity in high quality InSb 2DEGs and demonstrates the creation of robust, gate-tunable $\pi$-Josephson junctions. We show that the $0-\pi$ transition can be driven both by magnetic fields and gate voltages. The significant region of phase space where the $\pi$–JJ is stable could prove advantageous in the study of topological superconductivity in planar JJs[12,13,20,21]. Moreover, these large SOI 2DEGs, in conjunction with our magnetic field compatible superconducting electrodes and clear Landau quantization, would also be excellent candidates to realize topological junctions in the quantum Hall regime[7]. Finally, the ability to control the ground state between 0 and $\pi$ states using gates is analogous to recent experimental results in ferromagnetic JJs[43], and could possibly serve as a semiconductor-based platform for novel superconducting logic applications[44]. We therefore establish InSb 2DEGs as a new, scalable platform for developing hybrid superconductor-semiconductor technologies.

## Methods

**Wafer growth**. InSb-based 2DEGs were grown on semi-insulating GaAs (100) substrates by molecular beam epitaxy in a Veeco Gen 930 using ultra-high purity techniques and methods as described in ref. [45]. The layer stack of the hetero-structure is shown in Supplementary Fig. 1a. The growth has been initiated with a 100 nm thick GaAs buffer followed by a 1 μm thick AlSb nucleation layer. The metamorphic buffer is composed of a superlattice of 300 nm thick $In_{0.91}Al_{0.09}Sb$ and 200 nm thick $In_{0.75}Al_{0.25}Sb$ layers, repeated three times, and directly followed by a 2 μm thick $In_{0.91}Al_{0.09}Sb$ layer. The active region consists of a 30 nm thick InSb quantum well and a 40 nm thick $In_{0.91}Al_{0.09}Sb$ top barrier. The Si δ-doping layer has been introduced at 20 nm from the quantum well and the surface. The $In_xAl_{1-x}Sb$ buffer, the InSb quantum well and the $In_xAl_{1-x}Sb$ setback were grown at a temperature of 440 °C under a p(1 × 3) surface reconstruction. The growth temperature was lowered to 340 °C, where the surface reconstruction changed to c (4 × 4), just before the δ-doping layer, to facilitate Si incorporation[46]. The scanning transmission electron micrograph of Supplementary Fig. 1b reveals the efficiency of the metamorphic buffer to filter the dislocations.

**Device fabrication**. The devices are fabricated using electron beam lithography. First, mesa structures are defined by etching the InSb 2DEG in selected areas. We use a wet etch solution consisting of 560 ml deionized water, 9.6 g citric acid powder, 5 ml $H_2O_2$ and 7 ml $H_3PO_4$, and etch for 5 min, which results in an etch depth around 150 nm. This is followed by the deposition of superconducting contacts in an ATC 1800-V sputtering system. Before the deposition, we clean the InSb interfaces in an Ar plasma for 3 min (using a power of 100 W and a pressure of 5 mTorr). Subsequently, without breaking the vacuum, we sputter NbTi (30 s) and NbTiN (330 s) at a pressure of 2.5 mTorr, resulting in a layer thickness of approximately 200 nm. Next, a 45 nm thick layer of $AlO_x$ dielectric is added by atomic layer deposition at 105 °C, followed by a top-gate consisting of 10 nm/170 nm of Ti/Au.

## Data availability

All data files are available at 4TU.ResearchData repository, https://doi.org/10.4121/uuid:5fab8273-8794-4cd7-96d4-ba8ec00a62cf

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

## Acknowledgements

We thank Ady Stern, Attila Geresdi, and Michiel de Moor for useful discussions. The research at Delft was supported by the Dutch National Science Foundation (NWO) and a TKI grant of the Dutch topsectoren program. The work at Purdue was funded by Microsoft Quantum.

## Author contributions

C.T.K. and C.M.M. fabricated and measured the devices. C.T., G.C.G. and M.J.M. designed and grew the semiconductor heterostructures. C.T., S.M., C.R.G., R.K., T.W., R.E.D., G.C.G. and M.J.M. characterized the materials. M.L. and G.S. provided the effective mass measurements. C.T.K., C.M.M., F.K.d.V. and S.G. performed the data analysis. The manuscript was written by C.T.K., F.K.d.V., C.M.M. and S.G., with input from all co-authors. S.G. supervised the project.

## Additional information

**Competing interests:** The authors declare no competing interests.

