## [Peer Review File · Nature Communications]

Reviewers' comments:

Reviewer #1 (Remarks to the Author):

The manuscript reports on proximity-induced superconductivity and tunable pi-junctions formed in InSb quantum wells. While the results are of interest to publish in some form after revision, it is not recommended for publication in Nature Communications. The main comment is that the claim for providing the first evidence of induced superconductivity in such InSb quantum wells is not valid, since similar results have been reported in e.g., Ref. 3 in such material systems and other quantum well systems. Here, in this manuscript, tunable 0-pi transitions using in-plane magnetic field, gate voltage, and junction length are studied in more details. While these results are worth for publication in a special journal after revision, it does not meet requirements for publication in Nature Communications. Following are my comments/questions on the contents of the manuscript for improvement.

- The behaviors of $I_{sR_n} - 1/L$ (or $/L^2$) relations. In particular, for ballistic junctions, I_{sR_n} should not have such a behavior. For example, at L goes to zero, $1/L$ dependence indicates that I_{sR_n} goes infinity, which I do not see it is physically reasonable.

- The schematic picture showing in the inset of Fig. 2(a) needs theoretical justification. Application of an in-plane field may make band to become helical bands, but I could not see as it is so simple that the Fermi circles are rigidly shifted.

- The critical in-plane field at which I_c revivals is not small and thus it could lead to magnetic depopulation of carriers in the quantum wells. This is because the magnetic length at the revival field is comparable to the thickness of the quantum wells. Magnetic-field orientation dependent measurements at the normal states should be carried out to confirm that the depopulation effect of in-plane magnetic field is negligible in the work.

- In Fig. S4, it is not explained why at zero in-plane field, the Interference pattern is not symmetric with respect to zero out-plane field and why the interference pattern is not at all symmetric with respect the white lines drawn in the figure for finite in-plane field.

Overall I feel that too many unclear questions remain in the manuscript. Thus, the authors may have to make effort on a deep understanding of their results.

Below I also have a technical comment, namely,

- There are unacceptable many typos in the reference list, which makes the reviewer hard to locate the dedicated references.

Reviewer #2 (Remarks to the Author):

This manuscript reports on inducing superconductivity in high mobility InSb 2DEGs. The field of semiconductor-superconductors has experienced significant growth since the report of Majorana zero-modes in nanowires, but is still in search of a material platform that will enable networks of ballistic 1D superconducting electronics. While InSb nanowires are not a scalable platform, InSb 2DEGs have shown much promise (eg, with high mobilities), but the difficulty in nanofabrication on MBE-grown InSb 2DEGs has slowed progress. The field would benefit from the advancement of fabricating

superconducting devices in MBE-grown InSb 2DEGS.

In this paper, ballistic superconducting features are reported in InSb 2DEGs Josephson junctions (JJs). The authors report on a $1/L$ scaling in the long junction limit of the $I_c R N$ up to micron-scale junctions. This $1/L$ dependence in the long junction has been theoretically known since Ishii in 1970 and has only been demonstrated in the micron-scale in h-BN encapsulated graphene JJs. Such a demonstration highlights the extremely low levels of disorder in MBE grown InSb 2DEGs and potential for engineering ballistic superconducting networks.

Of particular importance, the authors convincingly demonstrate a ballistic n junction, with control over the 0 - n transition tuned with length, magnetic field, and density. This transition is relevant for studying Majorana modes and topological superconductivity, especially at relatively low magnetic fields.

Overall, this is an excellent manuscript, with clear, convincing, important, and timely data. I support publication in Nature Communications, but would like to see a few small items addressed:

1. While the paper focused on the long junction limit, the transparency of these junctions is relatively low. While this detail is not important for the $1/L$ scaling of $I_c R N$, a brief mention of processes that can achieve higher transparency, such as epitaxial semiconductor-superconductor techniques that are heavily used in the field, would be worthwhile (cf, Zhang, H. et al. Ballistic superconductivity in semiconductor nanowires. Nat. Commun. 8, 16025 (2017) and Gill, S. T. et al. Selective-area superconductivity to ballistic semiconductor nanowires. 2018 Nano Lett. 18 6121).
2. While $I_c R N$ is generally independent of gate, I would like to know (maybe in the supporting information) under what conditions they extracted the $I_c R N$ values.
3. Do these junctions pinch off? The low-density regime is probably the most favorable for topological superconductivity applications, and I wonder why there is no data below -0.5 V. Also, do the authors have an estimate of the number of occupied subbands in these 2DEGs?
4. Do the authors have additional data for short-limit junctions? I think that discussing the short limit to long junction crossover would be a valuable addition to the supporting information.

Reviewer #3 (Remarks to the Author):

Ke and co-workers report the fabrication and magnetoelectric characterization of InSb 2DEG-NbTiN hybrid Josephson junctions, providing the first evidence of induced superconductivity in high-quality InSb quantum wells. The junctions show Josephson coupling over several microns thereby supporting the frame of the ballistic transport regime.

Moreover, the large g -factor peculiar of this semiconductor 2DEG joined with the gate control of charge carriers in the weak-links allow to tune the current-phase relation of the junctions, enabling the transition from the 0 to the n state in a well-defined way. In particular, the author show that such a 0 - n transition can be driven by both gate voltages and magnetic fields applied in the plane of the 2DEG. The authors claim that the present results suggest that InSb 2DEGs could be a promising platform for investigating the interplay between magnetism, spin-orbit interaction and proximity-induced superconductivity.

The paper is clearly and well written, figures are nice and well constructed, the physical results seem solid, and the bibliography appears to be complete. To my mind the paper is suitable for publication in Nature Communications. Before this, I ask the author to comment the following issues:

i) The Josephson junctions magnetic characterization has been performed with an in-plane static field directed along the y axis, i.e., perpendicular to the transport direction. I would be interesting a comment on the behavior of the Josephson coupling and current-phase relation versus magnetic field applied along the x axis. What is to be expected? A comment for the reader would be valuable.

ii) Figure 1b shows a different Fraunhofer pattern for positive and negative currents flowing through the junction. What is the reason for this behavior? Please explain.

iii) Up to which temperature was the Josephson coupling observed in these junctions?

iv) Did the author performed any characterization of the Josephson critical current as a function of bath temperature? A comment on this question would be important.

The reviewers comments are in black, our response in blue, and changes to the manuscript are indicated in red.

Reviewer #1 (Remarks to the Author):

The manuscript reports on proximity-induced superconductivity and tunable pi-junctions formed in InSb quantum wells. While the results are of interest to publish in some form after revision, it is not recommended for publication in Nature Communications. The main comment is that the claim for providing the first evidence of induced superconductivity in such InSb quantum wells is not valid, since similar results have been reported in e.g., Ref. 3 in such material systems and other quantum well systems. While the results are of interest to publish in some form after revision, it is not recommended for publication in Nature Communications.

Ref 3 in the original manuscript [Pribyl et al, Nature Nano (2015)] deals with *InAs quantum wells*. In contrast, our study deals with *high mobility InSb quantum wells*. The study of induced superconductivity in InAs quantum wells dates back to the 1980s [see for example: Takayanagi; PRL 54, 2449 (1985)], with impressive developments more recently in the form of epitaxially grown superconductors on InAs [Shabani; PRB 93, 155402 (2016) and Kjaergaard; Nat. Comm 7, 12841 (2016)]. It is therefore not our intention to (nor do we in the manuscript) claim that we are the first to induce superconductivity in semiconductor quantum wells. Our focus is on InSb quantum wells, a different material system. In the context of topological superconductivity, they are expected to be superior, mainly because of the large g-factor (~ 25) compared to As based systems ($\sim 3-6$ in InGaAs, 8-10 in InAs). Superconductivity in InSb quantum wells have however not been studied extensively yet. The only result in literature to our knowledge is Ref. 10 in the original manuscript, where a low mobility (therefore low quality) InSb layer is used. We, in contrast, present superconducting transport through high mobility (high quality) InSb quantum wells. The high quality is signified by the ballistic superconducting transport found in our devices over micron scale distances. We would thus have to stand by our claim that this is the first demonstration of induced superconductivity in high mobility InSb 2DEGs, as also recognized by Reviewer 2 and 3.

Next, we would like to address the comment that “similar results have been reported in Ref3 and other quantum well systems”. In Ref 3 the authors studied interference patterns in Josephson junctions (JJs) as a function of gate voltage to demonstrate edge-mode superconductivity, possibly arising from the special inverted band structure in these quantum wells. Neither does Ref 3 deal with InSb quantum wells (as explained above), nor with the response of JJs to a Zeeman field ($0-\pi$ transitions), nor does it demonstrate any ballistic transport. Therefore, it is not completely clear to us how these measurements are similar to ours, beyond the mere observation of the Josephson effect.

In summary, we believe that the novelty of our work goes beyond the observation of induced superconductivity in a new material system (i.e., high mobility InSb 2DEGs), but actually exploits the exceptional properties of our InSb JJs to perform a set of experiments which have thus far not been reported in other semiconductor quantum wells.

The behaviors of $I_{sR_n} - 1/L$ (or $/L^2$) relations. In particular, for ballistic junctions, I_{sR_n} should not have such a behavior. For example, at L goes to zero, $1/L$ dependence indicates that I_{sR_n} goes infinity, which I do not see it is physically reasonable.

These scaling arguments were developed theoretically many decades ago [Altshuler et al., (Ref 33 of original manuscript) and Ishii et al [Progress of Theoretical Physics 44, 1525 (1970)]. This is also recognized by the comment from Reviewer 2: “This $1/L$ dependence in the long junction has been

theoretically known since Ishii in 1970 and has only been demonstrated in the micron-scale in h-BN encapsulated graphene JJs.”

As we have noted in the manuscript, these results are valid in the long junction limit ($L > \xi$), where L is the length of the JJ, and ξ is the induced superconducting coherence length. Therefore, by definition it cannot be extrapolated to arbitrarily small values of L . In contrast, when $L \ll \xi$ (not relevant for the JJs studied in this work), junctions are described by the formalism of the “short-junction limit” where $I_c R_n$ does not depend on the Thouless energy any more, but is only determined by the superconducting gap [Beenakker & van Houten, Phys. Rev. Lett. 66, 3056 (1991)]. Therefore, this scaling in the long junction limit is justified.

For clarity we have now also included the reference from Ishii et al. (ref 35 in the revised manuscript), when we discuss these scaling arguments in the manuscript.

The schematic picture showing in the inset of Fig. 2(a) needs theoretical justification. Application of an in-plane field may make band to become helical bands, but I could not see as it is so simple that the Fermi circles are rigidly shifted.

We realize that these aspects could be made clearer in our description of the Fermi surfaces. This picture is theoretically justified as long as SOI dominates over the Zeeman energy. To clarify the schematic further: solid circles are Fermi surfaces at $B = 0$, and spin orbit coupling governs the spin-orientation, giving rise to spin-momentum locking in both the bands. The effect of magnetic field is not to “make these bands helical” (or open a helical gap, as may be the case in one-dimensional systems), but only to shift the Fermi surfaces as described in theory (for example in Ref 14, 17 and 20 in original manuscript). Furthermore, Fig S3 and associated text describes how in our system the largest energy is indeed Δ_{SO} , which makes this picture valid.

We have moved the text in the footnote (Ref 37) into the main text, which states “The depicted shift of the Fermi surfaces assumes that the spin-orbit energy dominates over the Zeeman energy, which is indeed the case for the measured JJs (see SI for a detailed discussion).” Also, we now reference 14, 17 and 20 to direct the reader to the appropriate literature about large SOI 2DEGs in Zeeman fields.

The critical in-plane field at which I_c revivals is not small and thus it could lead to magnetic depopulation of carriers in the quantum wells. This is because the magnetic length at the revival field is comparable to the thickness of the quantum wells. Magnetic-field orientation dependent measurements at the normal states should be carried out to confirm that the depopulation effect of in-plane magnetic field is negligible in the work.

This is an interesting point, which we had not considered in detail. Following the reviewers suggestion we have carried out further experiments of the normal state resistance (R_n).

The results are summarized in the Fig. R1. We have presented here additional data for the one of the JJs ($L = 1.1\mu\text{m}$) in the manuscript. To eliminate any remnant effects of superconductivity the measurements are performed at high temperature (4 K) and high DC current bias (90 μA). As seen in Fig R1 (a) the change in R_n with parallel magnetic field (B_y) is small (we checked this for $V_g = 0\text{V}$ and $V_g = -0.4\text{V}$). Figure R1 (b) shows that there is a small MR (a few %), however there is no correlation between the MR and the variation of I_s with B_y , which changes by more than an order of magnitude. For reference we also show in R1 (c) the relevant plot from the main text (i.e., Fig 2c).

Figure R1: In plane magnetoresistance (MR). (a) Normal state resistance (R_n) as a function of B_y for JJ ($L = 1.1$ μ m) at two different gate voltages. Calculated MR (in %). Variation of I_s with B_y for the same JJ at different gate voltages (Fig 2c in the manuscript).

In addition to the new data, it is worth pointing out a few other experimental observations which indicate that magnetic depopulation does not play a role in the modulation of critical current: (i) The range of densities explored in this study are limited to the regime where only a single sub-band is occupied in the quantum well, (ii) as the length of the JJ increases the revival field (Fig 2b in main text) decreases. If the revival was related to the magnetic length and quantum well width, one would have expected the transition to always occur at the same magnetic field irrespective of length (since the quantum well thickness is fixed for all devices). This is further supported by the gate dependence of the transition (Fig 3 in manuscript).

We have included this plot in the supplementary information in the section “In plane interference conditions”

5. In Fig. S4, it is not explained why at zero in-plane field, the Interference pattern is not symmetric with respect to zero out-plane field and why the interference pattern is not at all symmetric with respect the white lines drawn in the figure for finite in-plane field.

The asymmetry at larger magnetic fields has two possible origins which we expect to coexist in our samples. The first is the effect of magnetic vortices which nucleate in our type II superconductor (NbTiN) at higher magnetic fields. The second reason has to do with terms in the Hamiltonian that break mirror symmetry of the potential in the JJ (e.g., small amounts of disorder at the interface), as described in Rasmussen et al [PRB 93, 155406 (2016)]. To make sure that these effects do not influence our extraction of I_s , we performed two separate cooldowns (Fig 2a in manuscript) and confirmed that the results are in agreement. We attribute the small (but noticeable) changes between the cooldowns to slightly different vortex configurations in the superconducting leads. However, they do not affect the value of magnetic field at which the supercurrent revives.

The asymmetry seen in Fig. S4 at zero in plane field is related to the fact that this data was obtained after a large field had already been applied to the sample (resulting in vortices). Even when the in-plane field is removed there can be flux trapping in the leads causing small distortions in the interference pattern. In Fig R2 we show an example of the same device without any history of large magnetic fields and we indeed see that the pattern is symmetric, while the maximal supercurrent is hardly changed as compared to S4. We note that the offset (~ 22 mT) is a trivial one that is associated with an offset in the current source of our magnet.

To clarify this, we have now added a discussion about the asymmetries in the supplementary information section titled “magnetic field alignment”.

Figure R2: Symmetric Fraunhofer pattern obtained before application of large magnetic fields.

Overall I feel that too many unclear questions remain in the manuscript. Thus, the authors may have to make effort on a deep understanding of their results.

Hopefully we have adequately answered the questions listed by the referee.

There are unacceptable many typos in the reference list, which makes the reviewer hard to locate the dedicated references.

We have now corrected the typographical errors.

Reviewer #2 (Remarks to the Author):

This manuscript reports on inducing superconductivity in high mobility InSb 2DEGs. The field of semiconductor-superconductors has experienced significant growth since the report of Majorana zero-modes in nanowires, but is still in search of a material platform that will enable networks of ballistic 1D superconducting electronics. While InSb nanowires are not a scalable platform, InSb 2DEGs have shown much promise (eg. with high mobilities), but the difficulty in nanofabrication on MBE-grown InSb 2DEGs has slowed progress. The field would benefit from the advancement of fabricating superconducting devices in MBE-grown InSb 2DEGS.

We thank the reviewer for appreciating the relevance of high quality InSb 2DEGs as a scalable material platform for Majorana devices, and recognizing the existing challenges in device fabrication associated with them.

In this paper, ballistic superconducting features are reported in InSb 2DEGs Josephson junctions (JJs). The authors report on a $1/L$ scaling in the long junction limit of the IcRN up to micron-scale junctions. This $1/L$ dependence in the long junction has been theoretically known since Ishii in 1970 and has only been demonstrated in the micron-scale in h-BN encapsulated graphene JJs. Such a demonstration highlights the extremely low levels of disorder in MBE grown InSb 2DEGs and potential for engineering ballistic superconducting networks.

We are indeed not aware of any semiconductor quantum well where this prediction has been observed experimentally, and the phenomenological work of Ishii is particularly relevant in this regard.

To emphasize this, we have added the reference to this work [Ishii; Progress of Theoretical Physics 44, 1525 (1970)] in the revised manuscript (Ref. 35) .

Of particular importance, the authors convincingly demonstrate a ballistic π junction, with control over the $0-\pi$ transition tuned with length, magnetic field, and density. This transition is relevant for studying Majorana modes and topological superconductivity, especially at relatively low magnetic fields.

Overall, this is an excellent manuscript, with clear, convincing, important, and timely data. I support publication in Nature Communications, but would like to see a few small items addressed:

We thank the reviewer for the positive recommendation and recognizing its relevance to the ongoing work on topological superconductivity. Further comments are addressed below.

1. While the paper focused on the long junction limit, the transparency of these junctions is relatively low. While this detail is not important for the $1/L$ scaling of I_{cRN} , a brief mention of processes that can achieve higher transparency, such as epitaxial semiconductor-superconductor techniques that are heavily used in the field, would be worthwhile (cf, Zhang, H. et al. Ballistic superconductivity in semiconductor nanowires. Nat. Commun. 8, 16025 (2017) and Gill, S. T. et al. Selective-area superconductivity to ballistic semiconductor nanowires. 2018 Nano Lett. 18 6121).

Yes, our transparencies are low (0.6-0.7) compared to, for example, InAs/Al hybrid devices. One possibility to make this better could be to spend more effort on optimizing the NbTiN/InSb interface. But, as the reviewer suggests, it would be ideal to extend the techniques of Aluminum deposition on InSb nanowires to 2DEGs. In fact, we have been actively pursuing this. One issue that complicates InSb/Al hybrids is the chemical reaction between Al and InSb which creates a barrier layer at the interface, see for example: F. Boscherini; Phys. Rev. B 35, 9580 (1987). We have been working on strategies to mitigate this in order to create stable InSb/Al planar devices.

Following the reviewers suggestion we have now added a discussion (including the references cited above) about possible improvements in the supplementary section "Multiple Andreev reflections and excess current" where we have calculated the transparencies.

2. While I_{cRN} is generally independent of gate, I would like to know (maybe in the supporting information) under what conditions they extracted the I_{cRN} values.

These values were extracted at zero gate voltage (i.e., density = $2.7 \times 10^{11} \text{ cm}^{-2}$).

We have now added this information in the caption of Fig. 1 in the main text.

3. Do these junctions pinch off? The low-density regime is probably the most favorable for topological superconductivity applications, and I wonder why there is no data below -0.5 V. Also, do the authors have an estimate of the number of occupied subbands in these 2DEGs?

The junctions do pinch off, and we have checked this on several junctions in the past. During these experiments we did not want to push the gate to more negative values, since it was not essential to the study, and we wanted to avoid possible gate leakage. However, to definitively answer the

question of the reviewer, we have now re-measured one of the JJs in this study ($L = 1.1\mu\text{m}$) up to more negative gate voltages (Fig R3) and it does indeed pinch off without any gate leakage.

Figure R3: Complete pinch-off for Josephson junction. Raw data without any corrections for filter resistances in the fridge wiring

From measurements on Hall bars on the wafers used in these studies we are certain that we have only one occupied subband in the density range studied in this experiment. At higher density (around $3.5 \text{ e}11 \text{ cm}^{-2}$) we do see the appearance of a second sub-band.

4. Do the authors have additional data for short-limit junctions? I think that discussing the short limit to long junction crossover would be a valuable addition to the supporting information.

We have only measured a few devices with $L < 500 \text{ nm}$, but focused on the long junctions since the required magnetic fields for the $0-\pi$ transitions are significantly lower. It is therefore difficult at this point for us to make a comparison between short/long junctions. However, in the light of recent theoretical studies short junctions would likely be better suited to study topological superconductivity, since the topological gap is expected to scale as $1/L^2$ [Pientka; PRX 7, 021032 (2017)]. Getting to the extreme short junction limit is challenging using NbTiN, since it has a large superconducting gap. This is another reason (in addition to better interface transparencies) that Al/InSb junctions would be ideal to study JJs in the short junction limit.

Reviewer #3 (Remarks to the Author):

Ke and co-workers report the fabrication and magnetoelectric characterization of InSb 2DEG-NbTiN hybrid Josephson junctions, providing the first evidence of induced superconductivity in high-quality InSb quantum wells. The junctions show Josephson coupling over several microns thereby supporting the frame of the ballistic transport regime.

We thank the reviewer for appreciating the novelty of demonstrating ballistic superconductivity in high quality InSb 2DEGs.

Moreover, the large g-factor peculiar of this semiconductor 2DEG joined with the gate control of charge carriers in the weak-links allow to tune the current-phase relation of the junctions, enabling the transition from the 0 to the π state in a well-defined way. In particular, the author show that such a 0- π transition can be driven by both gate voltages and magnetic fields applied in the plane of the 2DEG. The authors claim that the present results suggest that InSb 2DEGs could be a promising platform for investigating the interplay between magnetism, spin-orbit interaction and proximity-induced superconductivity.

The paper is clearly and well written, figures are nice and well constructed, the physical results seem solid, and the bibliography appears to be complete. To my mind the paper is suitable for publication in Nature Communications. Before this, I ask the author to comment the following issues:

We thank the reviewer for the positive recommendation and recognizing the advantages of InSb 2DEGs as a material platform for hybrid devices. Further comments are addressed below.

i) The Josephson junctions magnetic characterization has been performed with an in-plane static field directed along the y axis, i.e., perpendicular to the transport direction. I would be interesting a comment on the behavior of the Josephson coupling and current-phase relation versus magnetic field applied along the x axis. What is to be expected? A comment for the reader would be valuable.

In the absence of any spin-orbit interaction (SOI) magnetic field along x and y directions should be equivalent since the transitions are only driven by a Zeeman field. However, in the presence of SOI there can be variations depending on the field orientation [see for example, Ref 14 of original manuscript: Bezuglyi; PRB 66, 1 (2002)]. However, it has been shown that geometry related effects such as flux focusing can strongly influence the dependence of critical current when the magnetic field is applied perpendicular to the leads (i.e. x-direction) completely masking any effects of SOI [Suominen; PRB 95, 035307 (2017)]. We also performed (limited) experiments with magnetic field along the x-axis and observed that the supercurrent falls much faster (presumably due to such geometric effects). At this point, without additional data, it would be difficult for us to comment further on exactly how the CPR changes.

We now make a comment about this in the main text: “The orientation of the magnetic field reflects the Fermi surfaces described and avoids unwanted geometric effects [Suominen; PRB 95, 035307 (2017).]”

ii) Figure 1b shows a different Fraunhofer pattern for positive and negative currents flowing through the junction. What is the reason for this behavior? Please explain.

This asymmetry is a manifestation of hysteresis in the IV curve. Sweeping the current bias from negative to positive (as is the case in all the data in this paper) shows a switch from the dissipative branch to the supercurrent branch (referred to as the ‘re-trapping current’, I_r). Upon increasing the bias to positive values the junction switches again at a value known as the ‘switching current’, $I_s > I_r$. Reversing the direction of the sweep results in the mirror image where I_r (I_s) occur at positive (negative) bias. These are typical characteristics of underdamped Josephson junctions described within the RCSJ model. However, it has been shown extensively in literature that for SNS junctions the dominant mechanism for this hysteresis (resulting in the asymmetry between positive/negative currents) is in fact related to self-heating and dissipation in the JJ. Starting from a high bias (when the junction is resistive) the power dissipation causes heating which effectively puts the junction out of

thermal equilibrium (hence the lower value of I_r). However, once in the zero-resistance branch there is no further dissipation resulting in $I_s > I_r$.

iii) Up to which temperature was the Josephson coupling observed in these junctions?

We have not performed many measurements at high temperatures on the specific junctions described in detail in the paper, but have done some experiments on other JJs on the same wafer. Fig R4 shows measurements at 1.87K for junctions of two different lengths. While the longer one shows indications of superconductivity (minimum in dV/dI), for shorter junctions superconductivity is fully developed ($dV/dI = 0$). At 4.2 K neither JJ showed any supercurrent.

Figure R4: Josephson junctions at higher temperatures

iv) Did the author performed any characterization of the Josephson critical current as a function of bath temperature? A comment on this question would be important.

Unfortunately, at this moment we have not done a detailed temperature dependence of these samples. In principle, comparing such T-dep data of I_c with numerics (the Eilenberger equation) could be used as additional evidence for ballistic superconductivity. However, we did not perform these experiments due the clear $1/L$ dependence (Fig 1d) combined with the quantitative agreement of the $0-\pi$ transitions with the ballistic Thouless energy (indicated in Fig 4 of the manuscript). Another interesting future study would be to study the temperature dependence of the I_s vs B curves (of Fig. 2). According to theory [Pientka; PRX 7, 021032 (2017)], as the temperature is increased the nodes in supercurrent should become much deeper since higher order terms in the current phase relation drop rapidly with temperature.

REVIEWERS' COMMENTS:

Reviewer #2 (Remarks to the Author):

I believe the authors have sufficiently and appropriately responded to my comments, as well as the critiques of the other reviewers. The manuscript is now suitable for publications.